# Lunasin Improves the LDL-C Lowering Efficacy of Simvastatin via Inhibiting PCSK9 Expression in Hepatocytes and ApoE^−/−^ Mice

**DOI:** 10.3390/molecules24224140

**Published:** 2019-11-15

**Authors:** Lili Gu, Yaqin Gong, Cheng Zhao, Yue Wang, Qinghua Tian, Gaoxin Lei, Yalin Liang, Wenfeng Zhao, Shuhua Tan

**Affiliations:** State Key Laboratory of Natural Medicines, Jiangsu Key Laboratory of Druggability of Biopharmaceuticals, School of Life Science and Technology, China Pharmaceutical University, Nanjing 210009, China; lyguli2008@163.com (L.G.); yaqin602@163.com (Y.G.); zc11235813@outlook.com (C.Z.); wangyuejn@163.com (Y.W.); hua13770992123@163.com (Q.T.); raystar-85@163.com (G.L.); cpu12444lyl@126.com (Y.L.); wenfengzhaocpu@126.com (W.Z.)

**Keywords:** lunasin, simvastatin, proprotein convertase subtilisin/kexin type 9, low density lipoprotein receptor, low density lipoprotein cholesterol, hepatocyte nuclear factor-1α

## Abstract

Statins are the most popular therapeutic drugs to lower plasma low density lipoprotein cholesterol (LDL-C) synthesis by competitively inhibiting hydroxyl-3-methyl-glutaryl-CoA (HMG-CoA) reductase and up-regulating the hepatic low density lipoprotein receptor (LDLR). However, the concomitant up-regulation of proprotein convertase subtilisin/kexin type 9 (PCSK9) by statin attenuates its cholesterol lowering efficacy. Lunasin, a soybean derived 43-amino acid polypeptide, has been previously shown to functionally enhance LDL uptake via down-regulating PCSK9 and up-regulating LDLR in hepatocytes and mice. Herein, we investigated the LDL-C lowering efficacy of simvastatin combined with lunasin. In HepG2 cells, after co-treatment with 1 μM simvastatin and 5 μM lunasin for 24 h, the up-regulation of PCSK9 by simvastatin was effectively counteracted by lunasin via down-regulating hepatocyte nuclear factor 1α (HNF-1α), and the functional LDL uptake was additively enhanced. Additionally, after combined therapy with simvastatin and lunasin for four weeks, ApoE^−/−^ mice had significantly lower PCSK9 and higher LDLR levels in hepatic tissues and remarkably reduced plasma concentrations of total cholesterol (TC) and LDL-C, as compared to each monotherapy. Conclusively, lunasin significantly improved the LDL-C lowering efficacy of simvastatin by counteracting simvastatin induced elevation of PCSK9 in hepatocytes and ApoE^−/−^ mice. Simvastatin combined with lunasin could be a novel regimen for hypercholesterolemia treatment.

## 1. Introduction

Elevated circulation cholesterol level, especially low density lipoprotein cholesterol (LDL-C), contributes to the main risk factors of cardiovascular disease (CVD). The National Lipid Association (NLA) recommended atherogenic cholesterols including non-high density lipoprotein cholesterol and LDL-C as the primary targets of cholesterol lowering therapies [1].

Statins, a class of specific hydroxy-3-methyl-glutaryl-CoA (HMG-CoA) reductase inhibitors, have been advised as the first-choice hypocholesterolemic agents by the American College of Cardiology (ACC)/American Heart Association (AHA) [2]. In principle, statins can suppress the synthesis of endogenous cholesterol by competitively inhibiting HMG-CoA reductase and accelerate clearance of circulation LDL-C by up-regulating hepatic low density lipoprotein receptor (LDLR) [3,4,5]. However, statins may strongly up-regulate proprotein convertase subtilisin/kexin type 9 (PCSK9) by inducing the expression of hepatocyte nuclear factor 1α (HNF-1α), the dominating transcription factor of PCSK9 to cause the resistance to the LDL-cholesterol lowering effect of statins [6,7]. The underlying mechanism is that the up-regulated PCSK9 reduces the LDLR level via binding to it and transporting it to the lysosome for degradation [8], which eventually attenuates the clinical performance of statins to a large extent. Accordingly, various statin combination therapy approaches have been considered to enhance the LDL-C lowering effect and reduce the dosage of statins, as well as decrease the risk of adverse effects in the clinic [9].

Combination therapy appears to be a solution to improve the LDL-C lowering efficacy of statins and circumvent issues of statin resistance and intolerance. Thus far, it has been reported that bile acid sequestrants (BAS) in combination with statin therapy provide additive reductions in LDL-C compared with statin monotherapy [10,11]. The cholesterol absorption inhibitor ezetimibe, when combined with a statin, lowers LDL-C more than statin alone [12]. Niacin plus statin offsets the increase in PCSK9 levels noted with statin therapy [13]. Simvastatin together with either carvacrol or berberine improves its lipid lowering efficacy [14,15].

Lunasin, a 43-amino acid polypeptide with a molecular weight of ~5 kDa, which was initially identified from soybean [16], has been previously proven to possess various pharmacological activities against cancer [17], inflammation [18], and CVD [19]. Interestingly, we previously revealed that lunasin can functionally enhance LDL uptake in hepatocytes via both inhibiting PCSK9 expression and enhancing LDLR level, thereby remarkably reducing total cholesterol (TC) and LDL-C in blood as compared to vehicle control [20]. Thus, in this study, we investigated whether simvastatin combined with lunasin could improve the cholesterol lowering efficacy of simvastatin.

## 2. Results

### 2.1. Lunasin Suppresses Simvastatin Induced Elevation of PCSK9 Levels via Down-Regulating HNF-1α in HepG2 Cells

Statin has been known to lower plasma LDL-C synthesis by competitively inhibiting HMG-CoA reductase and up-regulating the hepatic LDLR. However, the concomitant up-regulation of PCSK9 by statin promotes the degradation of LDLR and thereby attenuates its cholesterol lowering efficacy [21,22,23]. Thus, we examined whether the up-regulation of PCSK9 by simvastatin was inhibited by lunasin in HepG2 cells. As shown in Figure 1A,B, 1 μM simvastatin treatment significantly increased PCSK9 expression at the mRNA and protein levels, while 5 μM lunasin treatment remarkably inhibited PCSK9 expression at the mRNA and protein levels in HepG2 cells as compared to vehicle control. The combination treatment of simvastatin with lunasin reduced PCSK9 expression at the mRNA and protein levels as compared to simvastatin treatment alone. Thus, it was implied that lunasin significantly suppressed the simvastatin induced elevation of the PCSK9 level in HepG2 cells.

Further, the expression level of HNF-1α, a dominating regulator of PCSK9, was analyzed in HepG2 cells; as shown in Figure 1C,D, the HNF-1α expression was stimulated by simvastatin at the mRNA (Figure 1C) and protein (Figure 1D) levels. However, as compared to simvastatin treatment alone, combination treatment of lunasin with simvastatin effectively reduced the HNF-1α expression level at the mRNA and protein levels. We further investigated whether the down-regulation of PCSK9 by lunasin was mediated by HNF-1α. HepG2 cells were pre-treated with *HNF-1α* siRNA before the treatment of lunasin. Importantly, as shown in Figure 1E, F, knock-down of *HNF-1α* by siHNF-1α effectively abolished the up-regulation of HNF-1α or PCSK9 induced by simvastatin treatment; a similar tendency was also observed by simvastatin combined with lunasin. Taken together, it was demonstrated that lunasin counteracted simvastatin induced elevation of PCSK9 expression at least partially via down-regulating HNF-1α in HepG2 cells.

### 2.2. Simvastatin Combined with Lunasin Synergistically Increases LDLR Level and Functionally Enhances LDL Uptake in HepG2 Cells

To detect the effect of simvastatin combined with lunasin treatment on the LDLR level, HepG2 cells were treated with 1 μM simvastatin and/or 5 μM lunasin for 24 h immediately after a one hour depletion of serum with opti-minimum essential media (Opti-MEM) medium. Then, the LDLR mRNA and protein levels were determined by quantitative real-time PCR (qRT-PCR) and Western blot. It was shown that treatment with either simvastatin or lunasin alone significantly increased the LDLR mRNA and protein levels. Moreover, lunasin combined with simvastatin treatment additively increased the LDLR level as compared to either lunasin or simvastatin alone (Figure 2A,B). Beyond that, functional analysis indicated that lunasin plus simvastatin treatment exhibited additive enhancement in LDL uptake in HepG2 cells (Figure 2C).

### 2.3. Lunasin Reduces LDLR Degradation by Counteracting Simvastatin-Induced Up-Regulation of PCSK9 in ApoE^−/−^ Mice

ApoE^−/−^ mice fed a high fat diet (HFD) were administrated with simvastatin and/or lunasin on a daily basis. After four weeks of administration, we measured PCSK9 and LDLR levels in liver tissues of ApoE^−/−^ mice. As shown in Figure 3A,B, hepatic PCSK9 expression was dramatically up-regulated by simvastatin alone; however, it was significantly suppressed at both the mRNA and protein levels in the group treated by simvastatin in combination with lunasin. Besides, immunohistochemistry staining indicated that PCSK9 secreted in the liver of ApoE^−/−^ mice was apparently reduced in the lunasin added simvastatin group (Figure 3C,D). Furthermore, qRT-PCR and Western blot analysis showed that simvastatin stimulated up-regulation of hepatic HNF-1α was effectively counteracted by lunasin (Figure 3A,B).

Additionally, it was shown that hepatic LDLR mRNA and protein levels were elevated by administration with either simvastatin or lunasin alone as compared to the model control group (ApoE^−/−^ mice fed with HFD and i.p. administrated with vehicle); however, they were remarkably up-regulated to a greater extent when treated with simvastatin combined with lunasin (Figure 4A,B). The data indicated that simvastatin combined with lunasin could enhance LDLR expression level more effectively than simvastatin monotherapy via suppressing simvastatin induced elevation of the PCSK9 level in ApoE^−/−^ mice fed with HFD.

### 2.4. Simvastatin Combined with Lunasin Improves Its Serum Cholesterol Lowering Efficacy in ApoE^−/−^ Mice

Given that the synergistic effects of simvastatin plus lunasin on elevating the LDLR level and functionally enhancing LDL uptake were observed in HepG2 cells, we investigated the in vivo anti-hyperlipidemia activity of simvastatin combined with lunasin in ApoE^−/−^ mice. After four weeks of administration, the serum cholesterol concentrations were analyzed in ApoE^−/−^ mice fed an HFD, and it was found that simvastatin monotherapy failed at lowering LDL-C and TC concentrations relative to the model group; however, lunasin treatment alone effectively reduced the serum LDL-C and TC levels, and lunasin plus simvastatin showed more potent serum cholesterol lowering efficacy in ApoE^−/−^ mice than lunasin monotherapy (Figure 5A,B).

## 3. Discussion

LDL-C is not only involved in the formation of cardiovascular diseases, but also closely related to other chronic diseases. The levels of circulating LDL are directly associated with atherosclerosis disease severity, especially ox-LDL [24]. Accumulation of ox-LDL in liver resident macrophages contributes to inflammation and disease progression of non-alcoholic steatohepatitis (NASH) [25]. Several studies showed that levels of serum ox-LDL were increased in patients with breast, pancreas, colon, or esophageal cancer, and ox-LDL induced mutagenesis, stimulated proliferation, initiated metastasis, and induced treatment resistance [26,27]. It was observed that patients with impaired renal function exhibited altered lipid metabolism and dyslipidemia [28], which may contribute to the worsening renal function and to the development of cardiovascular complications [29,30]. It was shown that higher levels of total cholesterol and LDL-C were observed in rats with experimental chronic renal failure, which positively correlated with circulating PCSK9 and negatively with the levels of LDLR [31]. Thus, effectively lowering circulating LDL is essential for disease treatment.

Statins have been accepted as the first-line therapy for lowering LDL-C in the management of patients with increased risk for CVD and associated mortality; however, some patients treated with statins appear to be statin resistant because they fail to achieve adequate reduction of LDL-C levels, while others are statin intolerant because they are unable to tolerate statin therapy due to adverse effects, particularly myopathy and increased activity of liver enzymes [10].

We have previously revealed that lunasin treatment effectively inhibited PCSK9 expression and remarkably elevated LDLR level in hepatocytes and mice [20]; thus, we were prompted to explore whether combination therapy with simvastatin and lunasin could enhance the LDL-C lowering efficacy of simvastatin. In liver tissue, PCSK9 synthesis is largely controlled at the gene transcriptional level by HNF1; there are two members of the HNF1 family, HNF1α and HNF1β. Previous research identified a highly conserved HNF1 binding site on the PCSK9 promoter region as another critical regulatory sequence motif of PCSK9 transcription [32]. The importance of HNF1α in PCSK9 expression has been clearly demonstrated in cell culture studies and in mice where adenovirus-mediated overexpression of HNF1α led to increased PCSK9 and reduced liver LDLR protein [33]. As expected, it was found that lunasin effectively counteracted simvastatin induced elevation of PCSK9 by decreasing HNF-1α, thereby increasing the LDLR level and thus functionally enhancing LDL uptake in HepG2 cells.

Herein, we investigated the in vivo anti-hyperlipidemia activity of simvastatin combined with lunasin in ApoE^−/−^ mice fed an HFD. It was observed that simvastatin monotherapy had little effect on lowering serum LDL-C and TC concentrations in ApoE^−/−^ mice. This result was consistent with previous reports that the lipid lowering effect of statins depends on the presence of intact apolipoprotein E, which functions to transport circulating cholesterol into cells, particularly hepatocytes, and acts as an important mediator for hepatic metabolic clearance of circulating cholesterol [34,35]. Likewise, it was confirmed in the clinic that *ApoE* genotypes were associated with variations in plasma-lipid levels and with responses to statins [36,37], and the polymorphism in ApoE could cause statin resistance [38]. However, in this study, simvastatin combined with lunasin showed more potent serum cholesterol lowering efficacy in ApoE^−/−^ mice than each monotherapy (Figure 5A, B), indicating that lunasin could effectively reduce simvastatin resistance through counteracting simvastatin induced elevation of PCSK9.

## 4. Materials and Methods 

### 4.1. Materials

Simvastatin was obtained from Nanjing Duly Biotechnology Co., Ltd. (Nanjing, China). Lunasin was prepared in our laboratory by using recombinant DNA technology as previously described [39]. Dil-LDL was obtained from Yiyuan Biotechnologies (Guangzhou, China). Lipofectamine 3000 reagent, MEM, opti-MEM, and fetal bovine serum (FBS) were purchased from Thermo Fisher Scientific (Waltham, MA, USA). Anti-LDLR, anti-HNF-1α, and anti-PCSK9 antibodies were purchased from Abcam (Cambridge, UK). Anti-β-actin was obtained from Cell Signaling Technology (Danvers, MA, USA). Analysis kits for TC and LDL-C were purchased from Jiancheng Biotechnologies (Nanjing, China). The 4,6-diamidino-2-phenylindole (DAPI) dye was purchased from KeyGEN BioTECH (Nanjing, China). RNAiso plus reagent, the reverse transcription kit, and SYBR^®^ Premix Ex Taq ™II were obtained from Takara Bio (Shiga, Japan).

### 4.2. In Vivo Study

The protocol for the in vivo study was approved by the Animal Ethics Committees of China Pharmaceutical University (No. 201601179, 19 October 2016) and conformed to the Guide for the Care and Use of Laboratory Animals published by the National Institutes of Health.

Six-week-old male ApoE^−/−^ transgenic mice on a C57BL/6 background and their WT littermates were purchased from Beijing Vital River Laboratory Animal Technology Co., Ltd. (Beijing, China; No. SCXK 2012-0001) and maintained in an specific pathogen free(SPF)-class animal house at 25 °C, 40–60% humidity, with a 12 h light/dark cycle. Mice were provided with free access to food and water. The HFD containing 1.25% cholesterol (Diet^#^ D12108C, Research Diets, New Brunswick, NJ, USA) was used to induce hypercholesterolemia.

After an acclimatization period of 7 days, WT mice divided randomly into two groups (*n* = 8) were fed common chow and administrated with or without 0.5 μmol/kg of lunasin in an application volume of 0.1 mL/10 g body weight, while ApoE^−/−^ mice divided randomly into four groups (*n* = 8) were fed with HFD and administrated with 10 mg/kg simvastatin and/or 0.5 μmol/kg lunasin for 4 weeks, respectively. Lunasin was administrated by intraperitoneal injection, and simvastatin was given by oral gavage on a daily basis. Each animal was used only in one experiment in order to exclude the influence of other tests. At the end of the administration period, all mice were fasted for 8 h before blood sample collection and then euthanized for tissue harvest. All the experimental procedures were approved by the Animal Ethics Committee of China Pharmaceutical University.

### 4.3. Cell Culture and Treatments

HepG2 cells obtained from China Infrastructure of Cell Line Resources (Beijing, China) were cultured in MEM supplied with 10% (*V*/*V*) FBS, 100 units/mL penicillin, and 100 mg/mL streptomycin in a 37 °C, humidified incubator containing 5% CO_2_. Cells were seeded in six well plates and grown to 70% confluence followed by a one hour pretreatment with opti-MEM. Then, cells were treated with 1 μM simvastatin and/or 5 μM lunasin for 24 h, respectively. Cells treated with opti-MEM were used as the control. Total RNA and protein were extracted for qRT-PCR and Western blot analysis.

### 4.4. LDL Uptake Assay

The assay was conducted as described previously [40] with slight modification. Briefly, HepG2 cells were maintained in MEM supplemented with 10% FBS. The cells were seeded in 96 well black plates at a density of 1 × 10^4^ cells per well and grown to 70% confluence. Then, cells were incubated with serum-free opti-MEM for 1 h, followed by incubation with 1 μM simvastatin and/or 5 μM lunasin for 20 h. Thereafter, 20 μg/mL Dil-LDL were added, and the cells were incubated in the dark for an additional 4 h. Cells incubated with opti-MEM without Dil-LDL were used as the negative control. Cells incubated with Opti-MEM and 20 μg/mL Dil-LDL were used as the control for normalization, respectively. After rinsing with PBS 3 times, LDL uptake was measured on a fluorescence plate reader (Varioskan flash, Thermo, Waltham, MA, USA) at an excitation wavelength of 520 nm and an emission wavelength of 580 nm.

### 4.5. qRT-PCR

Total RNA was extracted from cells or liver tissue samples using RNAiso plus reagent and reverse transcribed into cDNA by a commercial reverse transcription kit. QRT-PCR was performed with 50~100 ng cDNA template and specific primers on an MX3000PTM qRT-PCR amplifier (Agilent Scientific, Palo Alto, CA, USA) using SYBR^®^ Premix Ex Taq ™II (Takara, Shiga, Japan) according to the manufacturer’s protocols. Primers for each gene are shown in Table 1. Target mRNA expression levels in each sample were normalized to the housekeeping gene *β-actin*. The 2^−ΔΔCt^ method was used to calculate relative mRNA expression levels [41]. Each run was completed with a melting curve analysis to confirm the specificity of amplification and lack of primer dimers. 

### 4.6. Western Blot

Cells or liver tissue samples were lysed or homogenized by using a glass homogenizer in cold RIPA lysis buffer (Solarbio, Beijing, China) containing 1 mM PMSF (Amresco, Solon, OH, USA), and the supernatant was collected. Total protein concentrations were determined by the BCA protein assay kit (Biouniquer, Beijing, China). An equal amount of proteins from each sample was separated by 10% SDS-PAGE and transferred onto a 0.22 μm PVDF membrane (Merck Millipore, Darmstadt, Germany) followed by blocking with TBS solution containing 0.1% (*V*/*V*) Tween 20 and 5% (*W*/*V*) nonfat milk for 1 h at room temperature. Then, the membrane was incubated with anti-PCSK9, anti-LDLR, anti-HNF-1α antibody, or anti-β-actin antibodies overnight at 4 °C with gentle shaking and subsequently incubated with a peroxidase-conjugated secondary antibody (1:5000) for 1 h at room temperature. Protein bands were developed by electrochemiluminescence (ECL) fluid (Thermo Scientific, Waltham, Massachusetts, USA) and quantified using Image J software.

### 4.7. Small Interference RNA Transfection

Pre-designed siRNAs targeted to human *HNF-1α* mRNA were forward: CAGUGAGA CUGCAGAAGUA, reverse: UACUUCUGCAGUCUCACUG, according to Luo’s study [42]. *HNF-1α* siRNAs and the negative control siRNA were synthesized by Shanghai Gene Pharmaceutical Technology Co. (Shanghai, China). The day before transfection, 2 × 10^5^ per well of HepG2 cells were plated in a 6 well plate with 2 mL MEM medium containing 10% FBS. When cells were grown to 70–80% confluence, cells were transfected with 100 pmol siRNA with 5 μL Lipofectamine 3000 according to the manufacturer’s instructions. Then, fresh growth medium was replaced 4 h after transfection. After 72 h transfection, cells were harvested for Western blot analyses. To examine whether gene silencing of *HNF-1α* affected the down-regulation of PCSK9 expression in HepG2 cells, 1 μM lunasin was added to treat the cells during the last 24 h of culture.

### 4.8. Serum Cholesterol Level Test

Blood samples were harvested from mice eyes and maintained at 4 °C for 4 h. Then, serum samples were obtained by centrifugation at 3000 rpm, at 4 °C, for 10 min, and the concentrations of LDL-C and TC were measured by using determination kits.

### 4.9. Immunohistochemistry Staining

For histological analysis, mice liver samples were fixed in 4% para-formaldehyde/PBS at 4 °C overnight and imbedded in paraffin followed by slicing on a microtome (RM2245, Leica, Germany) at a thickness of 5 μm. Liver sections were dewaxed and boiled in sodium citrate solution (0.01 M, pH 6.0) for 20 min and then treated with 3% H_2_O_2_ for 20 min at room temperature followed by blocking with 10% goat serum/TBST for 30 min. Subsequently, the sections were incubated with rabbit anti-PCSK9 polyclonal antibody (1:200) overnight at 4 °C, followed by incubation with Alexa Fluor^®^ 488 conjugated goat anti-rabbit IgG (1:400) as the secondary antibody for 1 h at room temperature. The cell nuclei were stained with DAPI. Finally, the sections were mounted by glycerin and photographed by Zeiss AX10 fluorescence microcopy (Zeiss, Oberkochen, Germany).

### 4.10. Statistical Analysis

The data are shown as the means ± SEM, and one-way ANOVA was performed using GraphPad Prism software 6.0 (San Diego, CA, USA). A value of *p* < 0.05 was considered statistically significant.

## 5. Conclusions

Conclusively, the present in vitro and in vivo study demonstrated that lunasin could effectively counteract simvastatin induced elevation of PCSK9 by decreasing HNF-1α, thereby synergistically increasing the LDLR level and functionally enhancing LDL uptake in hepatocytes, significantly improving the LDL-C lowering efficacy of simvastatin in ApoE^−/−^ mice (Figure 6). Simvastatin combined with lunasin could be a novel regimen for hypercholesterolemia treatment.

## Figures and Tables

**Figure 1 molecules-24-04140-f001:**
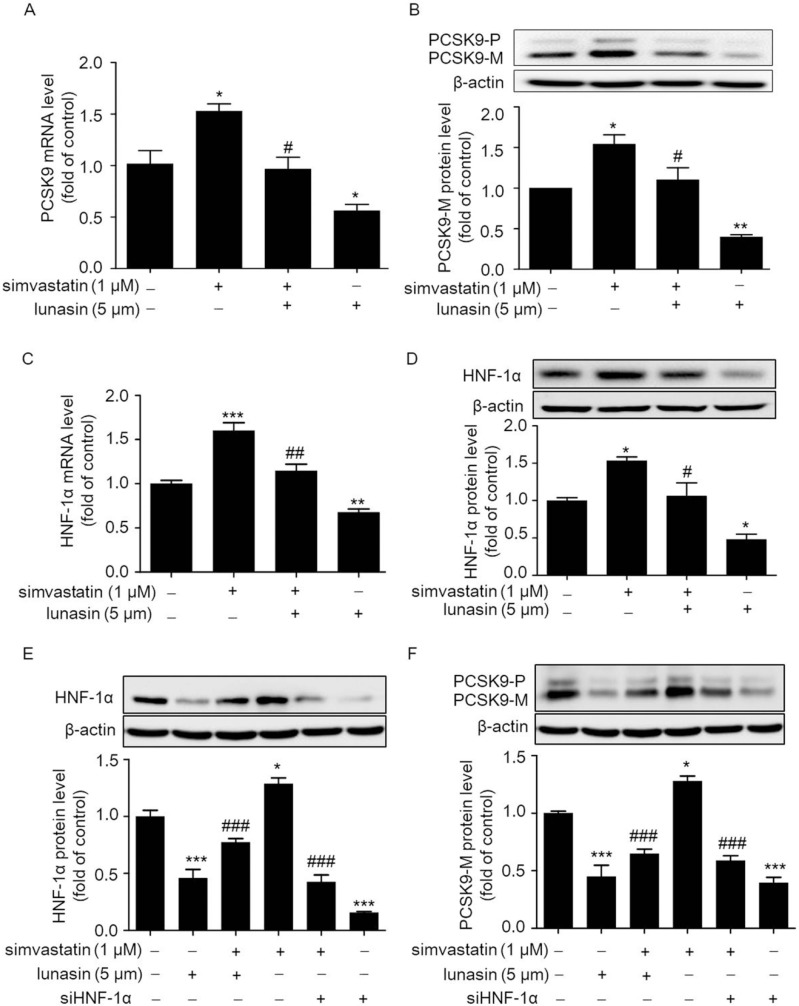
Effects of simvastatin combined with lunasin treatment on PCSK9 and HNF-1α expressions at the mRNA and protein levels in HepG2 cells. HepG2 cells were treated with simvastatin and/or lunasin for 24 h. The mRNA (**A**) and protein (**B**) levels of intracellular precursor PCSK9 (PCSK9-P) and mature PCSK9 (PCSK9-M), as well as the mRNA (**C**) and protein (**D**) level of HNF-1α were determined by qRT-PCR and Western blot using β-actin as an internal control, respectively. After transient transfection with siRNA for 4 h, EA.hy 926 cells were maintained in fresh medium for 48 h and treated with 1 μM lunasin for an additional 24 h. Then, the levels of HNF-1α (**E**) and PCSK9 (**F**) protein expression were analyzed by Western blot analyses, respectively. * *p* < 0.05, ** *p* < 0.01, *** *p* < 0.001 vs. the control group; ^#^
*p* < 0.05, ^##^
*p* < 0.01, ^###^
*p* < 0.001 vs. the simvastatin group (*n* = 3, means ± SEM).

**Figure 2 molecules-24-04140-f002:**
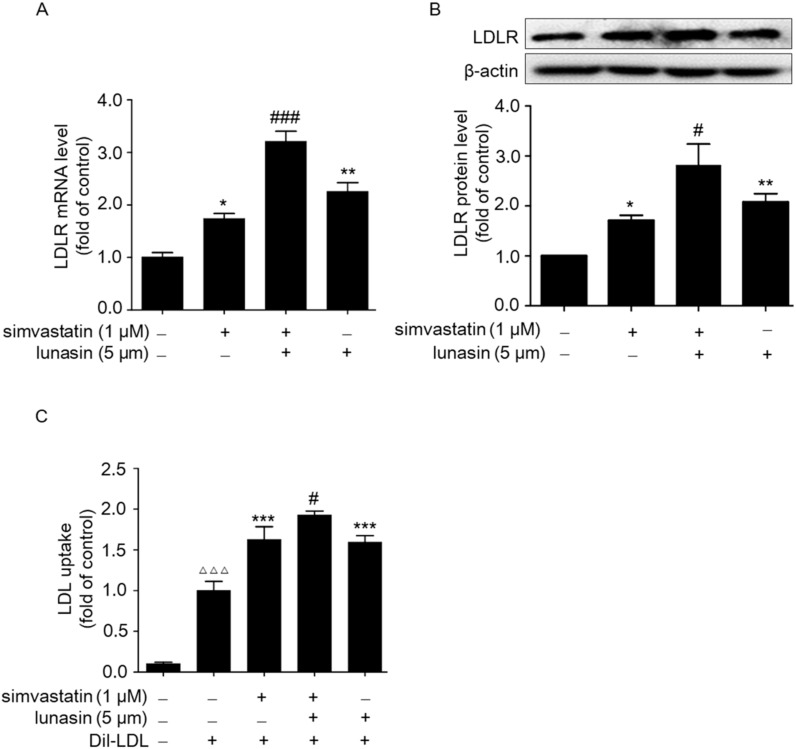
Effects of simvastatin in combination with lunasin treatment on the LDLR and LDL uptake levels in HepG2 cells. HepG2 cells were treated with simvastatin and/or lunasin for 24 h. The mRNA (**A**) and protein (**B**) levels of LDLR were analyzed by qRT-PCR and Western blot using β-actin as an internal control, respectively. * *p* < 0.05, ** *p* < 0.01 vs. the control group; ^#^
*p* < 0.05, ^###^
*p* < 0.001 vs. the simvastatin group. (**C**) LDL uptake was assessed in HepG2 cells after treatment with simvastatin and/or lunasin for 24 h on a fluorescence plate reader. ^ΔΔΔ^
*p* < 0.001 vs. the negativecontrol group; ^#^
*p* < 0.05 vs. the simvastatin group; *** *p* < 0.001 vs. the 20 μg/mL Dil-LDL group (*n* = 3, means ± SEM). Dil-DLD: LDL labeled with 1,1′-dioctadecyl-3,3,3′,3′-tetramethylindocarbocyanine perchlorate.

**Figure 3 molecules-24-04140-f003:**
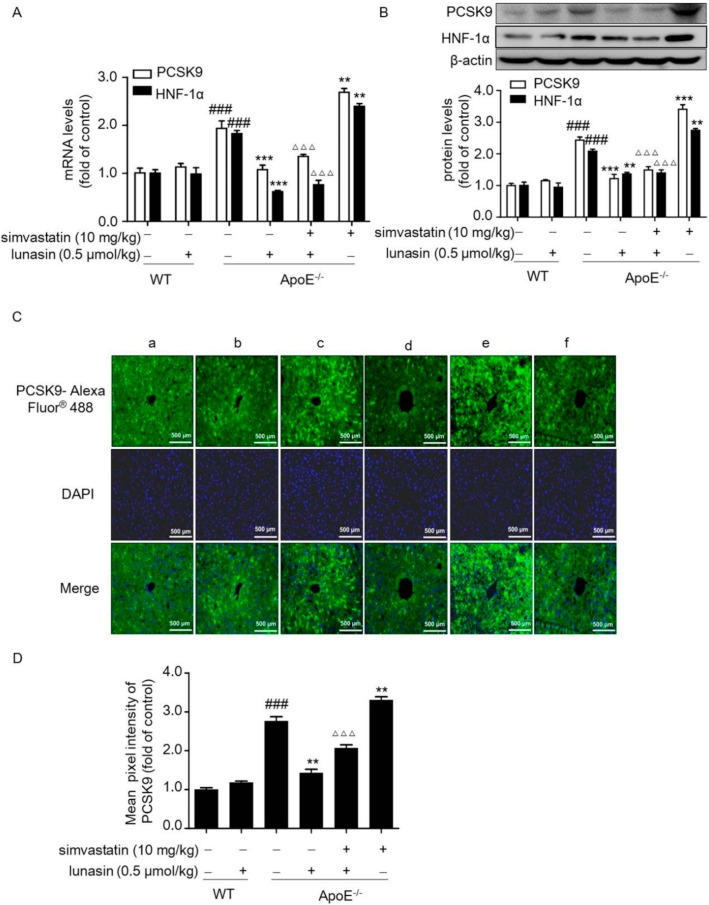
The combination of simvastatin with lunasin suppresses the up-regulation of PCSK9 induced by simvastatin in ApoE^−/−^ mice. ApoE^−/−^ mice were administrated with 10 mg/kg simvastatin and/or 0.5 μmol/kg lunasin on a daily basis for four weeks. The expression levels of PCSK9 andHNF-1α mRNA (**A**) and protein (**B**) in ApoE^−/−^ mice were determined by qRT-PCR and Western blot, respectively. The levels of PCSK9 secreted in hepatic tissues (**C**,**D**) were detected by immunohistochemistry staining (a: C57BL/6; b: C57BL/6 + 0.5 μmol/kg lunasin; c: ApoE^−/−^; d: ApoE^−/−^ + 0.5 μmol/kg lunasin; e: ApoE^−/−^ + 0.5 μmol/kg lunasin + 10 mg/kg simvastatin; f: ApoE^−/−^ + 10 mg/kg simvastatin). ^###^
*p* < 0.001 vs. C57BL/6 mice administrated with normal saline (NS); ** *p* < 0.01, *** *p* < 0.001 vs. ApoE^−/−^ mice administrated with NS; ^ΔΔΔ^
*p* < 0.001 vs. ApoE^−/−^ mice administrated with simvastatin (*n* = 8, means ± SEM). WT: wild type.

**Figure 4 molecules-24-04140-f004:**
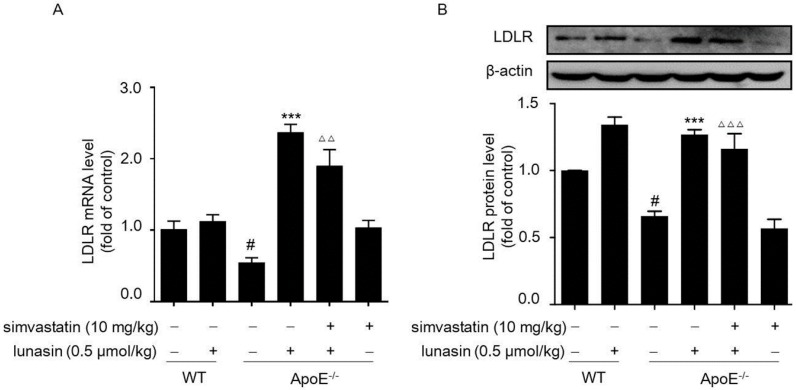
Effects of simvastatin in combination with lunasin on LDLR abundance in ApoE^−/−^ mice. ApoE^−/−^ mice were administrated with 10 mg/kg simvastatin and/or 0.5 μmol/kg lunasin on a daily basis for four weeks. The levels of LDLR mRNA (**A**) and protein (**B**) in liver tissues of ApoE^−/−^ mice were determined by qRT-PCR and Western blot, respectively. ^#^
*p* < 0.05 vs. C57BL/6 mice treated with normal saline (NS); *** *p* < 0.001 vs. ApoE^−/−^ mice treated with NS; ^ΔΔ^
*p* < 0.01, ^ΔΔΔ^
*p* < 0.001 vs. ApoE^−/−^ mice treated with simvastatin (*n* = 8, means ± SEM).

**Figure 5 molecules-24-04140-f005:**
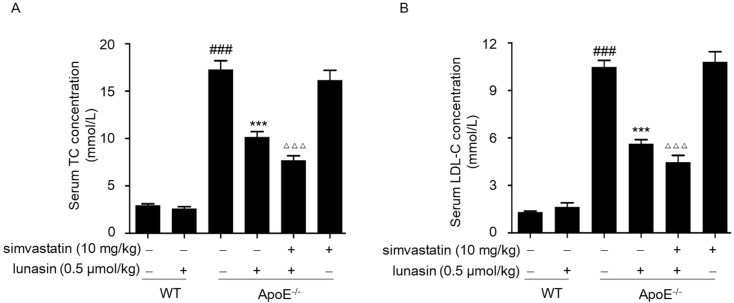
Effects of combined drug on serum cholesterol levels in ApoE^−/−^ mice fed with HFD. ApoE^−/−^ mice were i.p. administrated with 10 mg/kg simvastatin and/or 0.5 μmol/kg lunasin on daily basis for four weeks. LDL-C (**A**) and TC (**B**) concentrations in serum samples were measured by biochemical kits. ^###^
*p* < 0.001 vs. C57BL/6 mice administrated with normal saline (NS); *** *p* < 0.001 vs. ApoE^−/−^ mice administrated with NS; ^ΔΔΔ^
*p* < 0.001 vs. ApoE^−/−^ mice administrated with simvastatin (*n* = 8, means ± SEM).

**Figure 6 molecules-24-04140-f006:**
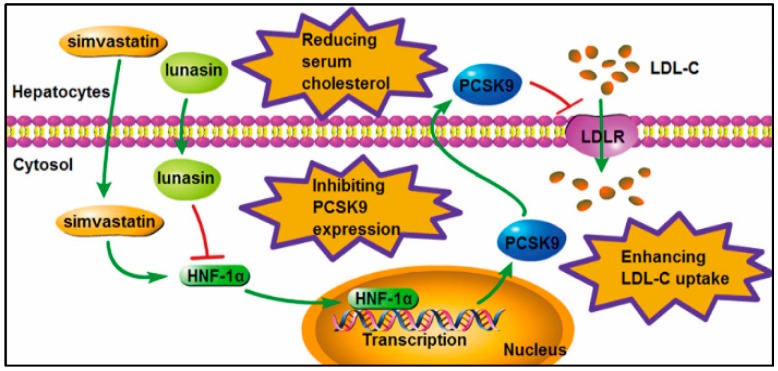
Schematic diagram of the molecular mechanism by which lunasin enhances the LDL-C lowering efficacy of simvastatin via down-regulating PCSK9 expression.

**Table 1 molecules-24-04140-t001:** Primer sequences used for qRT-PCR.

Gene	Forward Primer (5ʹ→3ʹ)	Reverse Primer (5ʹ→3ʹ)
*Homo-PCSK9*	AGGGGAGGACATCATTGGTG	CAGGTTGGGGGTCAGTACC
*Homo-HNF-1α*	AGGACGAGACGGACGACGAT	AGTGCCCTTGTTGAGGTGTT
*Homo-LDLR*	GAACCCATCAAAGAGTGCG	TCTTCCTGACCTCGTGCC
*Homo-β-actin*	CTCTTCCAGCCTTCCTTCCT	CAGGGCAGTGATCTCCTTCT
*Mus-PCSK9*	CAGAGGTCATCACAGTCGGG	GGGGCAAAGAGATCCACACA
*Mus-HNF-1α*	GAGCCTGAATCGAGCAGAAC	AGCCTTCTCTGGACACCTGA
*Mus-LDLR*	GTATGAGGTTCCTGTCCATC	CCTCTGTGGTCTTCTGGTAG
*Mus-β-actin*	GTGACGTTGACATCCGTAAAGA	GCCGGACTCATCGTACTCC

*Homo*: human; Mus: mouse *sapiens*; *PCSK9*: proprotein convertase subtilisin/kexin type 9; *HNF-1α:* hepatocyte nuclear factor 1α; *LDLR*: low density lipoprotein receptor.

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
