# Peer review of "Lunasin Improves the LDL-C Lowering Efficacy of Simvastatin via Inhibiting PCSK9 Expression in Hepatocytes and ApoE−/− Mice"

_molecules, 2019, doi:10.3390/molecules24224140_

Round 1

Reviewer 1 Report

The authors of this work report that Lunasin, a polypeptide present in soybean and a number of cereal grains, potenitates the LDLC-lowering effect of simvastatin in both hepatocytes as well as ApoE-deficient mice. The authors ascribe this effect of Lunasin to downregulation of HNF-1α and thus PCSK9 levels. This is an extension of the author’s previous work, reported in Oncotarget 2017, that potentially provides a rationale for further research on the suitabilitity of Lunasin as an add-on therapy to statins.

Comments:

Lines 107-110 (results section, 2.2): The authors claim that Lunasin exerted a synergistic effect on simvastatin-induced expression of LDLR mRNA and protein as well as LDL uptake. Judging from the graphs in figure 2, it appears that the effect of Lunasin on simvastatin is perhaps additive but not synergistic. Have the authors performed the necessary calculations to examine whether the observed effects are synergistic? Figure 2 shows that Lunasin improves the LDLR-inducing effect of simvastatin on LDLR mRNA and protein levels. The authors should discuss whether they believe this effect of Lunasin is solely mediated by its ability to abrogate the simvastatin-mediate induction of PCSK9 or by a combination of different mechanisms. In my opinion and given the author’s published results (Oncotarget 2017), the authors should consider the possibility that the potentiating effect of Lunasin on simvastatin-mediated induction of LDLR is mediated by not only the inhibitory effect of Lunasin on HNF-1α levels, but also by its stimulating effect of SREBP-2.

Author Response

We thanks for your good suggestion.

We have exchanged synergistic to additive in the revised manuscript in Lines 107-110 (results section, 2.2).

Statin, a class of specific hydroxy-3-methyl-glutaryl-CoA (HMG-CoA) reductase inhibitors, they depletes the intracellular cholesterol pool, which mobilizes the intracellular proteolytic processing machinery to release ER-bound SREBP2. The processed mature form of SREBP2 enters the nucleus where it binds to the SRE-1 element of the PCSK9 promoter and activates transcription. Because of the coexistence of an SRE-1 motif in the LDLR and PCSK9 promoters, statin treatment leads to increased transcription of both LDLR and its natural inhibitor, PCSK9[1, 2].The promoter region of the PCSK9 gene contains a sterol-regulatory element (SRE) site which can be transcriptionally activated by SREBP-2 and HNF-1α [3-5], HNF-1α plays a critical role in PCSK9 gene transcription and regulation as HNF1 site mutation reduced PCSK9 promoter activity > 90% [3]. Thus, the suppression of HNF-1α by lunasin predominantly inhibited the expression of PCSK9 in spite of the up-regulation of SREBP-2 by simvastatin.

Mayne, J.; Dewpura, T.; Raymond, A.; Cousins, M.; Chaplin, A.; Lahey, K. A.; Lahaye, S. A.; Mbikay, M.; Ooi, T. C.; Chrétien, M., Plasma PCSK9 levels are significantly modified by statins and fibrates in humans. 2008, 7, (1), 22-22. Careskey, H. E.; Davis, R. A.; Alborn, W. E.; Troutt, J. S.; Konrad, R. J., Atorvastatin increases human serum levels of proprotein convertase subtilisin kexin type 9 (PCSK9). Journal of Lipid Research 2008, 49, (2), 394-398. Li, H.; Dong, B.; Park, S. W.; Lee, H.-S.; Chen, W.; Liu, J., Hepatocyte Nuclear Factor 1? Plays a Critical Role in PCSK9 Gene Transcription and Regulation by the Natural Hypocholesterolemic Compound Berberine. Journal of Biological Chemistry 284, (42), 28885-28895. Jeong, H. J.; Lee, H.-S.; Kim, K.-S.; Kim, Y.-K.; Yoon, D.; Park, S. W., Sterol-dependent regulation of proprotein convertase subtilisin/kexin type 9 expression by sterol-regulatory element binding protein-2. Journal of Lipid Research 49, (2), 399-409. Dong, B.; Wu, M.; Li, H.; Kraemer, F. B.; Adeli, K.; Seidah, N. G.; Park, S. W.; Liu, J., Strong induction of PCSK9 gene expression through HNF1? and SREBP2: mechanism for the resistance to LDL-cholesterol lowering effect of statins in dyslipidemic hamsters. Journal of Lipid Research 51, (6), 1486-1495.

Reviewer 2 Report

This manuscript investigates a new therapy to lower plasma low-density lipoprotein cholesterol synthesis (LDL-C) in individuals with hypercholesterolemia. While previous statin-based therapies reduce LDL-C, the up-regulation of proprotein convertase subtilisin/kexin type 9 (PCSK9) can limit the effectiveness of statin-based therapies. The authors show that lunasin could be effective in combination with statins as a treatment for high cholesterol levels. Given the dangers of hypercholesterolemia, I find this manuscript to be both interesting and relevant.

I have a few questions and suggestions below:

While the authors apply an ANOVA statistical test to distinguish between control groups and varying data sets, they rarely compare between the different sample conditions. Providing an additional test between the combined statin and lunasin tests vs. the lunasin alone would provide interesting information regarding the efficacy of lunasin alone. Acronyms should be defined in the main text at the first point they are used. Some acronyms, such as HFD and TC, are specified in the methods section but not in the main text, which can lead to some confusion for readers if they approach the main text before the methods section. It would be helpful if the authors indicated what dosage of statin has been used in previous therapies to treat hypercholesterolemia. Presumably the dosage tested here is similar to what has been used previously in therapy with statin only? Do the authors have data on the overall impact of the combined treatment of statin and lunasin? Do the mice remain healthy throughout the treatment? Are there any potential side effects of the treatment observed?

Author Response

We thanks for your good suggestions.

We have previously revealed that lunasin can functionally enhance LDL uptake in hepatocytes via both inhibiting PCSK9 expression and enhancing LDLR level[1]. In this work, we focused on whether simvastatin combined with lunasin could improve the cholesterol lowering efficacy of simvastatin, so it is not very necessary to compare the efficacy of lunasin alone and simvastatin combined with lunasin.

We already carefully checked acronyms, and added full names in the revised manuscript.

According to previous research[2, 3], ApoE−/− mice were fed high fat diet (HFD) and treated with simvastatin (10 mg/kg per day), then in this study, we chose 10 mg/kg as the dosage of simvastatin.

The change of visceral index is a good way to judge thetoxic and effects of drugs on organs, we detected whether the drug affected the liver and renal index of mice. As shown in Figure. S1A, the liver index of mice was dramatically up-regulated in ApoE−/− mice fed HFD as compared to C57BL/6 mice administrated with normal saline, simvastatin monotherapy failed in lowering the liver index of mice. However, it was significantly suppressed in the group treated by simvastatin in combination with lunasin. There were no significant changes in renal index of mice after treatment by all kinds of drugs (Figure. S1B). Other potential side effects of the treatment were not observed in this study.

Figure S1. Effects of combined drugs on the visceral index of mice. ApoE-/- mice were i.p. administrated with 10 mg/kg simvastatin and/or 0.5 μmol/kg lunasin on daily basis for 4 weeks. The liver index (A) and renal index (B) of mice were measured. #### p < 0.0001 vs. C57BL/6 mice administrated with normal saline (NS); *p < 0.05 vs. ApoE-/- mice administrated with NS; Δ p < 0.05 vs. ApoE-/- mice administrated with simvastatin (n = 8, means ± SEM).

Gu, L.; Wang, Y.; Xu, Y.; Tian, Q.; Lei, G.; Zhao, C.; Gao, Z.; Pan, Q.; Zhao, W.; Nong, L., Lunasin functionally enhances LDL uptake via inhibiting PCSK9 and enhancing LDLR expression in vitro and in vivo. 2017, 8, (46), 80826-80840. Wang, Y.; Zhao, X.; Wang, Y.-S.; Song, S.-L.; Liang, H.; Ji, A.-G., An extract from medical leech improve the function of endothelial cells in vitro and attenuates atherosclerosis in ApoE null mice by reducing macrophages in the lesions. Biochemical & Biophysical Research Communications 455, (1-2), 119-125. Sang, H.; Yuan, N.; Yao, S.; Li, F.; Wang, J.; Fang, Y.; Qin, S., Inhibitory effect of the combination therapy of simvastatin and pinocembrin on atherosclerosis in apoE-deficient mice. Lipids in Health & Disease 11, (1).166-175
